# A Sustainability Innovation Experiential Learning Model for Virtual Reality Chemistry Laboratory: An Empirical Study with PLS-SEM and IPMA

**Chung-Ho Su \*** and **Ting-Wen Cheng**

Department of Animation and Game Design, Shu-Te University, Kaohsiung 824, Taiwan; ny88080@gmail.com
\* Correspondence: mic6033@stu.edu.tw; Tel.: +886-917-231-831

**Abstract:** This study focuses on serious virtual reality games, and how users can increase their understanding of the sustainable innovation learning (SIL) model and their familiarity with sustainable development strategies. "Users" of serious games consists of all possible target groups that are interested in attaining knowledge of sustainability through the use of games that are designed for a purpose beyond entertainment, in this case, for sustainable education. This research investigates the sustainable innovation experiential learning model by using a virtual chemistry laboratory to affect academic achievement. A questionnaire was completed by students who had used the virtual lab, and structural equation modeling (SEM) was applied for analysis. Importance-performance matrix analysis (IPMA) was able to help expand the basic partial least square (PLS)-SEM result with the fraction. The results show that experiential learning significantly affects learning motivation and academic achievement. Cognitive load and self-efficacy significantly affect learning motivation. Involvement significantly affects academic achievement. The virtual chemistry laboratory significantly affects academic achievement. Students who used the sustainability innovation experiential learning model obtained a better understanding of the chemical concepts. Moreover, a virtual lab promotes students' motivation in regard to chemistry.

**Keywords:** sustainability innovation learning; VR-environment; experiential learning; learning motivation; academic achievement; serious games

## 1. Introduction

The sustainable innovation learning model has become an indispensable learning quality for existing and future social citizens. The key issue in processing sustainable development indicators lies in reaching the balanced integration of a sustainable environment, sustainable society, and sustainable economy. However, it is not easy to achieve this goal, as the ultimate goal is a kind of responsible sustainable behavior. Auxiliary tools and learning strategies should be taken into consideration for supporting and promoting the quality of sustainable learning. Learning through gamification virtual reality can be a way to promote the sustainable innovation learning model. As stated in the literature, driven by the rising of costs of sustainable development as a result of environmental awareness and pollution, sustainable technologies have come a long way. [1–4]. The most promising achievements involve renewable energy, sustainable living, organic agriculture, environmental economics, and environmental technologies. Among such general categories, $CO^2$ capture, ocean cleanup, soil improvement, sustainable design, and green building technologies have increased the knowledge base of sustainability achievements. [5–8]. Nevertheless, as more attention is given to the demands for sustainability, it is commonly believed that reversal is required for learning knowledge through the sustainable innovation learning (SIL) model, as well as the promotion of

new teaching methods and educational tools [9–12]. In addition, various methods for innovative learning are being developed towards this direction. More and more primary and secondary education institutions are using and integrating SIL policies and sustainable communication models in their educational research [13]. In this study, education for sustainable innovation learning (ESIL) was regarded as meeting sustainability requirements, and thus, worthy of study. Gamification learning is a means of encouraging the concept of sustainable learning in society. [14].

These virtual worlds enable users to interact often with each other in an enjoyable and realistic environment; they can reflect the user's own experiences in daily life [15]. Virtual reality technology has more use than the traditional learning environment of chalk and blackboard. Individuals can interact or communicate with others through the systems or software, thus, it signifies great progress in the field of education. The virtual chemical laboratory is significantly effective in education, as it can promote learners to actively learn themselves and avoid the higher cost of real experiments [16–18]. A past study suggests that different fields of learning should identify whether other learning systems can cause positive outcomes [19]. As it is difficult for students to understand the concepts of chemistry, they must learn them through experiments [18]. Moreover, compared to virtual experiments, the costs of real experiments are much higher and require space where they can take place, as well as human and laboratory scheduling and security [20].

Over recent decades, primary and secondary education institutions have been researching how to incorporate sustainable development into their teaching methods [13,21,22]. More and more primary and secondary education institutions have developed innovative teaching tools for SIL, and carried out research on sustainable and integrated development, school partnership, institutional programs, traditional teaching programs of educators, and green campus programs [23]. In the innovative learning environment in universities, engineering-related higher education institutions (HEIs) in particular are taking basic and necessary steps to follow the SIL strategy to redevelop educational strategies, which has become a modern trend [24,25]. In respect of practical implications, it indicates that practitioners and educators can obtain in-depth understanding of existing gamification learning in relation to sustainability, which can provide the academic community with the learning reversal to further analyze and study the application of the SIL model in gamification learning.

Our study contributes to the field of virtual reality chemistry laboratory simulations learning for instructional use in experiential learning methods. This may pose different learning outcomes, because simulations and games have different design features and interactive models. Other researchers have also contributed different virtual simulated gamse as a student learning material, such as: Vfrog$^{TM}$ (Tactus Technologies, Akron, NY, USA), in which students dissect a virtual frog for learning biology [26]; and a virtual reality modeling language environment to teach high school students mathematical concepts for learning the mathematics logic and principles to solve a problem [27]. This study not only designed and developed a virtual reality chemistry laboratory simulation game, but also proposes a sustainability innovation experiential learning model to validate the learning effectiveness. This paper focuses on the effect of cognitive loading, and through the use of a virtual chemical laboratory, observes whether experiential learning, cognitive load, and self-efficacy affect learning motivation. The purposes of this paper are:

(1) To design and develop a virtual reality chemistry laboratory simulation game.
(2) To investigate the effects of cognitive load on academic achievement.
(3) To investigate the effects of self-efficacy on learning motivation.
(4) To investigate the effects of learning motivation on academic achievement.

## 2. Background

### 2.1. Virtual Reality in Science

Virtual reality (VR) is a technology based on computer programs to create a synthetic reality using 3D graphics. Virtual reality environments (VREs) allow individuals to interact simultaneously

with different elements and experience the presence of the environment. Virtual reality is interactive because users are not limited to being passive observers of the environment; rather, they can interact with different objects found in VREs, and the system will immediately respond to their actions. Virtual reality involves immersion because users can exist in a virtual world assisted by some equipment [28,29]. Therefore, virtual reality is defined as immersive and interactive, and relies on the visual, auditory, and tactile senses of computer media manipulation. This provides learners with a simulated experience of the world created by the computer program. From the user experience perspective, researchers see virtual reality as a psychological concept rather than a technical variable. Due to a combination of psychological and technical factors, virtual reality has been investigated in relation to several dimensions, such as presence, interaction, virtual identification, anonymity, synchronization, and three dimensional degrees [19,30]. Burdea and Coiffet [31] defined virtual reality as I$^3$: Immersion, Interaction, and Imagination. Immersion comprises mental and physical immersion; it plays an important role in the virtual world and creates successful personal experiences when using the systems to browse and control objects to achieve physical immersion in a human-made environment. In addition, users can see the changes on the screen according to their instructions. Learners can see and manipulate the objects on the screen, as well as also through their human senses. Users can gather information through virtual, auditory, and tactile means. Educators use virtual reality technology to promote learning activities. Interactions require detecting a user's instructions and responses in real time. Virtual reality is particularly helpful when dealing with problems requiring high creativity and problem-solving capabilities.

In recent years, more and more researches are focusing on virtual reality technology. Presence in a virtual world may turn on emotions in the real world [32]. Virtual reality is effective for practicing, and learning outcomes are significantly positively statistically correlated when combined with other teaching methods. Students perform better through learning in virtual reality than in a teacher-controlled learning environment, thus, education based on virtual reality is effective [33]. Virtual reality is widely used in academic research, as shown in Table 1, where earlier research on virtual reality applications is displayed.

**Table 1.** Earlier research on virtual reality (VR)**.**

| Field | Describing the Research | Conclusion | Researchers |
|---|---|---|---|
| Human-Computers Studies | A dataflow oriented development system for virtual reality applications | The simple matching system was developed in four versions on a virtual reality platform. The studies show how users interact with various versions. | [34] |
| Computers & Education | Learning science in a virtual reality application: The impacts of animated-virtual actors' visual complexity | There are positive correlations among presence, perceived emotion quality and learning perspective. There are no positive correlations with perceived difficulties, retention, or transfer. | [35] |
| Computers & Education | Educational virtual environments: A ten-year review of empirical research (1999–2009) | Collaboration and social negotiations are not only limited to participation in educational virtual environments, but also exist between participants and avatars, which provides a new dimension in computer assisted learning. | [36] |
| Human in Behaviors | Investigation of effects of virtual reality environments on learning performance of technical skills | Six modules: students prefer the simulation process and practical exercise among the six modules. | [37,38] |

**Table 1.** *Cont.*

| Field | Describing the Research | Conclusion | Researchers |
|---|---|---|---|
| Computers & Education | Effectiveness of virtual reality-based instruction on students' learning outcomes in K-12 and higher education: A meta-analysis | Mentioned that, in designing virtual reality instructions, the principles of instructional design should be considered. | [33] |
| Human in Behaviors | This study applied a virtual training environment to train police personnel for complex collaborative tasks. | In real and complex situations, through the measurement of learning transformation, virtual training is a good standard of training. | [39,40] |
| Human in Behaviors | Virtual reality and stimulation of touch and smell for inducing relaxation: A randomized controlled trial | When using a virtual reality environment, the sense of touch can improve efficacy because it provides more sensory information. | [41] |

### 2.2. Virtual Chemical Laboratory

Students learn the experimental process applied to the basic theory. One of the problems that students face in the classroom is associating concepts and physical models [18]. The learning environment can improve the students' interest in science [16]. When students go to the lab, it is not necessary to assess a new device or obtain new understanding of the world; they go to a laboratory to study what engineers already know.

Virtual experimentation is intended to be a separate section linked to other relevant areas of the site. Combining different methods with this approach can facilitate the process of learning and instruction in chemical engineering laboratories [42]. Martin-Villalba et al. [17] explained that a virtual lab is an effective educational tool, which provides a flexible and personalized approach to define the experiment performed in the model, and allows users to design and execute their own simulation experiments. The result is that users are actively engaged in their self-learning process. Virtual Lab is designed to provide users with the means for dynamic behavior in a convenient visual interaction model. The main purpose of the virtual lab is to provide traditional laboratory network support, thereby enhancing students' efficacy and independence in data analysis and report writing during experimental courses. Despite the fact that virtual laboratory experiments are not intended to replace real experiments, they reduce the costs and time required by experiments, due to errors leading to repeated experiments [18,20].

Virtual laboratories are being used more and more frequently and replacing real laboratories in biochemical engineering, as most of the students think they are useful, and the teacher can assess the performance of students and verify that they improved in the lab. This demonstrates that the integration of real experiments and virtual computer simulations can improve the learning process, as they are a complementary way to help students learn more efficiently and actively [16]. FeiselL and Rosa [43] compared the performance of a simulation laboratory and a traditional laboratory, and the results showed that students using the simulation laboratory received higher grades on written tests. Moreover, students' performance in using a simulation laboratory before a conventional laboratory is relatively good. The development of the virtual laboratory has great potential to enable students to achieve successful experiments, which cannot be completed or are difficult to achieve in a real laboratory [18].

### 3. Research Hypothesis

#### 3.1. The Relationships among Experiential Learning, Learning Motivation, and Academic Achievement

The experiential learning theory is a learning process of knowledge creation. Experiential learning is defined as knowledge created through the transformation of experiences, with the knowledge

resulting from a combination of understanding and experience conversion [44]. Experiential interactive learning environments often use Kolb experiential learning, which is a four-stage learning cycle concept [45]. As shown in Figure 1, the four stages of Kolb's experiential learning are: (1) concrete experience abilities (CE); (2) reflective observation abilities (RO); (3) abstract conceptualization abilities (AC); and (4) active experimentation abilities (AE). In particular, these four stages can be divided into two dimensions. The first dimension represents the concrete experiencing of events at one end and abstract conceptualization at the other. The other dimension has active experimentation at one extreme and reflective observation at the other.

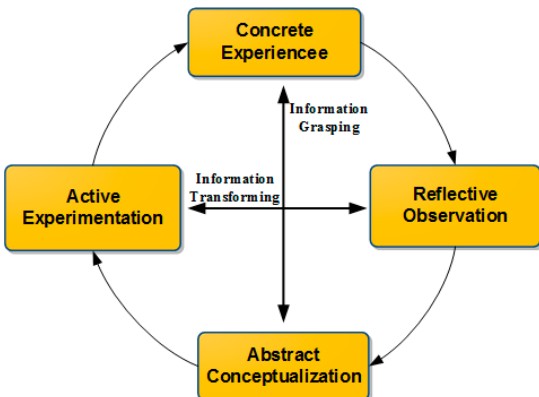

**Figure 1.** Kolb's experiential learning theory.

D. Kolb [45] concluded that learning only occurs when learners contact the information first, rather than through concrete experience or abstract concepts, and then, convert the information, meaning not through active experience or reflective observation. D. Kolb [45] also stated that transformative learning occurs at the intersection of concrete experience and abstract concepts. Reflecting on daily experience is one of the learning methods to improve one's knowledge or experience. Reflection is a necessary condition for turning experience into learning, and makes one question the validity and usefulness of experiences.

In the field of education, motivation can be defined as students wanting to work hard in the learning environment. Keller [46] proposed the attention, relevance, confidence, satisfaction (ARCS) model, which is comprised of four main elements. Attention: an element of motivation and also the prerequisite for learning. Relevance: many course designers and instructors make the course relate to students or to their future career opportunities. Confidence: there are many factors that affect a person's confidence; confident people believe that they can be successful. Satisfaction: resentment may occur when a teacher takes a student's reward or withholds incentives.

According to the above factors, the ARCS model causes individuals to pay attention, develop an interest, and find relationships. Next, it gives individuals the confidence to achieve their goals. Constructivists advocate that individuals learn from experiencing through the process of knowledge construction when learners complete meaningful tasks [47]. Experiential learning links learning motivation and academic achievement though immersion in virtual reality. Wu, Yan, Kao, Wang, and Wu [48] improved students' learning through the integration of a role-playing game (RPG) and the experiential learning cycle. The results showed that learning outcomes improved dramatically for students using RPGs. A previous study also pointed out that students participating in experiential learning have positive effects [49]. To sum up, two hypotheses are offered, as follows:

**H1:** *Experiential learning will significantly affect academic achievement.*

**H2:** *Experiential learning will significantly affect learning motivation.*

### 3.2. The Relationship between Cognitive Load and Learning Motivation

Most related research and empirical studies acknowledge that psychologists formulated the cognitive load theory and information process theory in the mid-20th century. The science of human behavior became a recognized discipline, which consisted of an empirical model: an experimental study design for interactions between humans and their environment [50].

The concentration of the cognitive load theory and cognitive resources is related to their use in the process of learning and problem-solving [51]. The cognitive load theory can be classified into three types [52]: intrinsic cognitive load is the load caused by the complexity of the material, which requires more memory when the intrinsic load is higher. The learning environments, problems, and tasks of intrinsic load are related to learners; this is a variable according to learners' previous experience in a field. While the intrinsic load cannot be decreased, learning tasks can be made more suitable for the learner's level of knowledge [53]. Therefore, segmenting the materials can decrease learning load and lead to better learning outcomes. The second type is extraneous load, which explains the results of the learning material design. Extraneous cognitive load is due to the characteristics of the learning situation, meaning it does not need the learning content, and may even hinder learning. Extraneous load is entirely related to the design of the new information presentation or learning experience. The third type is Germane cognitive load, which involves various processes, such as interpreting, exemplifying, classifying, inferring, differentiating, and organizing. The load that is imposed by these processes is denominated by the germane cognitive load. Pastore [53] described the finding that students under a high variability schedule report higher cognitive load and achieve better scores on transfer tests.

In the field of education, researchers believe that learning cognition is related to motivation. Students who are strongly interested in their learning tasks will pay more attention to them, as compared to students with lower interest. Students with lower motivation will exhibit less cognition, which reflects lower cognitive ability; conversely, higher motivation reflects higher cognitive ability [54–56]. Therefore, the following hypothesis is presented:

**H3:** *The cognitive load will significantly affect learning motivation.*

### 3.3. The Relationship between Self-Efficacy and Learning Motivation

Self-efficacy is defined as a person's judgment of his/her organization skills and the effectiveness of executing specified tasks [57]. Self-efficacy is a subjective judgment determined by a person who executes some actions, thus, it may not be able to accurately judge actual capability [55]. Self-efficacy provides a person with basic motivation and personal accomplishment, because when people believe they are able to achieve the expected results, they will take action regarding the difficulties and problems that they face [58]. Self-efficacy is an important factor of personal enhancement, and increasing self-efficacy can improve one's success regarding long-term challenges. A high level of self-efficacy and perceived control can improve efficiency and reduce the stressful state of emotional pain [59]. Typically, people tend to learn more and witness their achievements, thus self-efficacy has a significant impact on motivation, achievement, and learning [58].

Researchers believe that students' self-efficacy is related to their motivation, performance, and achievement [55]. Previous studies have confirmed that students' self-efficacy is related to different degrees of motivation. Bandura examined self-efficacy, and stated that an individual's ability is decided by whether his/her thoughts are positive or not; the more that they are positive, the more people can handle their difficulties and stimulate students' motivation for learning [60,61]. Student's self-efficacy has been proven to be positively correlated with intrinsic motivation. Student motivation is a process that affects achievement through self-efficacy, thus the following hypothesis is offered:

**H4:** *Self-efficacy will significantly affect learning motivation.*

*3.4. The Relationship between Learning Motivation and Academic Achievement*

Several researchers mentioned that academic achievement is an important construct, as academic achievement is as an educational outcome [55]. Academic achievement is defined as the degree that students achieve their educational goals. Typically, it involves achieving specific results in online assignments and tests, and is usually presented as grade point average (GPA) or a numeric rank [56]. There are two representative ways to investigate academic achievement: the first is the quantitative method, which is based on students' GPA. The other way is using a knowledge acquisition and achievement approach to extract the abstract factors [55]. Some studies suggest that, in a social expectation report, students may overestimate their own performance, particularly those in the lower grades, which can affect the validity of the results. Future research should use actual students' online results, rather than the online scores reported by students, in order to eliminate social bias [54].

Previous research analyzed the relationship between learning motivation (i.e., internal and external motivation) and GPA, as equated to academic achievement [55]. A cross-lagged regression model of academic motivation and achievement in high school was used to analyze the population interrelationships, and found that autonomic motivation, which defines relative autonomy in the past year, even after controlling for baseline achievement, was positively correlated with academic achievement [62]. This illustrated that the concept of learning motivation must be considered as a significant predictive effect of academic achievement. Their study suggested that, the more students are engaged in the enjoyment and challenge of learning motivation, the higher their academic achievement [55]. Therefore, another hypothesis is, as follows:

**H5:** *Learning motivation will significantly affect academic achievement.*

## 4. The VR Chemistry Laboratory Framework

*4.1. Framework of System Design*

According to the Game-based Learning Model developed by Garris, Ahlers, and Driskell [63], the process combines it with the learning motivation theory, cognitive load theory, and self-efficacy theory as the main feedback sustainability innovation experiential learning systems. This system includes three processes and a feedback system (Figure 2). Figure 2 shows the learning content, as based on experiential learning applied to the game design. The first part is External Process (Input), which is a gamified learning material design combined with virtual reality and experiential learning. It is designed through the 3D virtual gaming interface to enable players to control their learning cognitive loads more easily. The gamified integrated material design is also designed for reducing learning anxiety. The second part is Internal Process (Process), which sets gamification as the core design, and integrates learning motivation theory, cognitive load theory, and the learning motivation as the mental cycle learning mode to reinforce learning. The third part is the Output process (Output), which mainly processes the learning evaluation and feedback mechanisms. The system can be upgraded through the complete feedback cycle and correcting mechanism to reach the perfect instructional design. Table 2 shows the learning content based on experiential learning applied to game design.

**Table 2.** The learning content based on experiential learning applied to game design.

| Experiential Learning | The Chemical Experiment | The Level of Game Design | Learning Indicators |
|---|---|---|---|
| 1. concrete experience abilities | Experimental safety explanation | 1. Experimental safety explanation level | 1. Obtain the analysis of argument through the experimental results |
| 2. reflective observation abilities | Experimental procedures | 2. experimental level | 2. Observe their meaning and concept formation through the changes |
| 3. abstract conceptualization abilities | Principles explanation | 3. test level | 3. Using scientific terms, symbols, and common expressions correctly |
| 4. active experimentation abilities | | | 4. Understanding the changes through the collection of gases. |

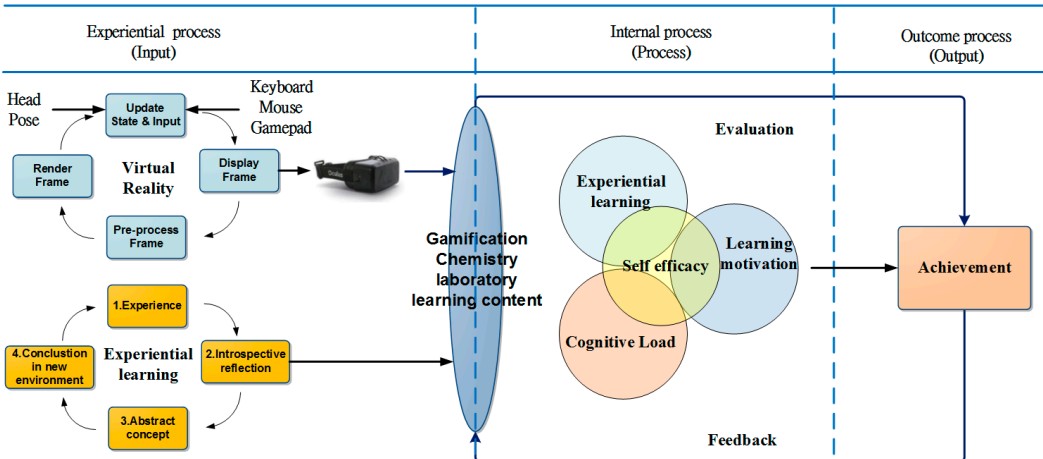

**Figure 2.** Sustainability innovation experiential learning performance system

*4.2. System Interface*

The gamification experiential learning performance system interface unit design mainly involves the experimental process of $CO_2$ gas collection, which is divided into 11 experimental steps. When learners complete the $CO_2$ collection experiment process, they will receive one badge and complete the solutions of the day. Students operate the virtual chemical laboratory; first, they receive an overview of the virtual chemical laboratory (as shown in Figure 3), where they collect the equipment they need for the collection process according to the experiment points (as shown in Figures 4 and 5), and complete the steps for collecting the $CO_2$ gas. At the same time, students interact with the virtual reality game and answer the learning questions. If students give a wrong answer, the system will go back to the screen with information hints. After students read related information, they can challenge the game again until they complete all the experimental steps. After students complete this stage, they will receive a password and coin to accumulate achievements and badges, which can allow learners to get to the next level of the chemical experiments.

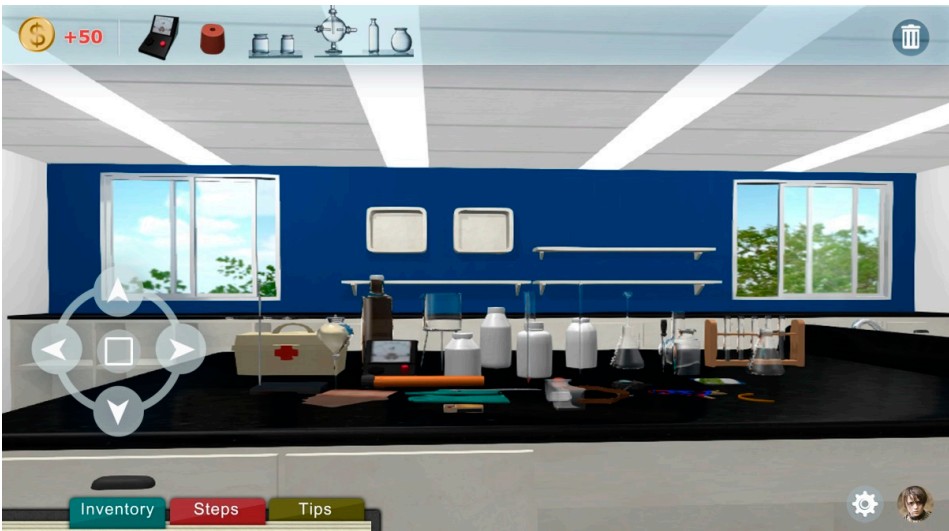

**Figure 3.** The overview of the virtual chemical laboratory.

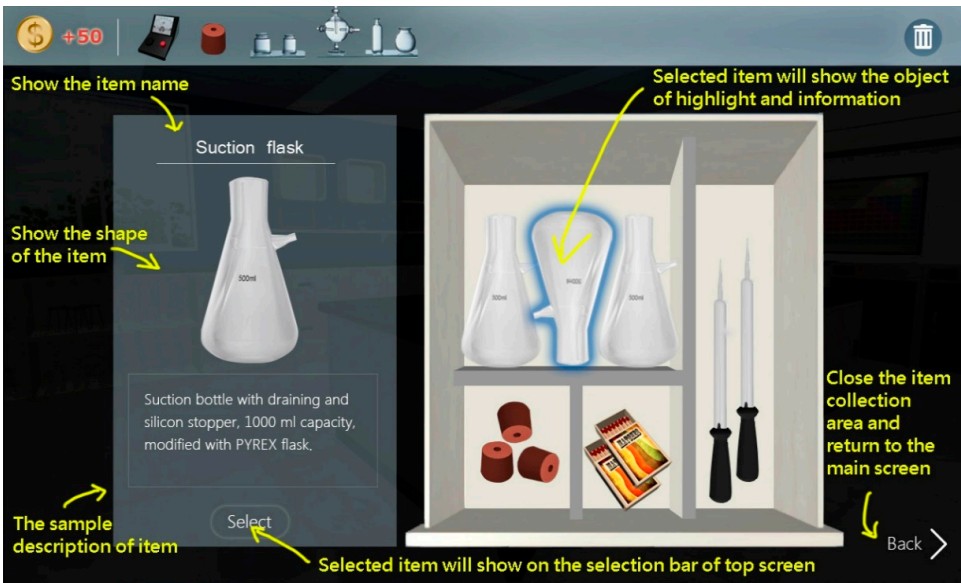

**Figure 4.** The equipment collection.

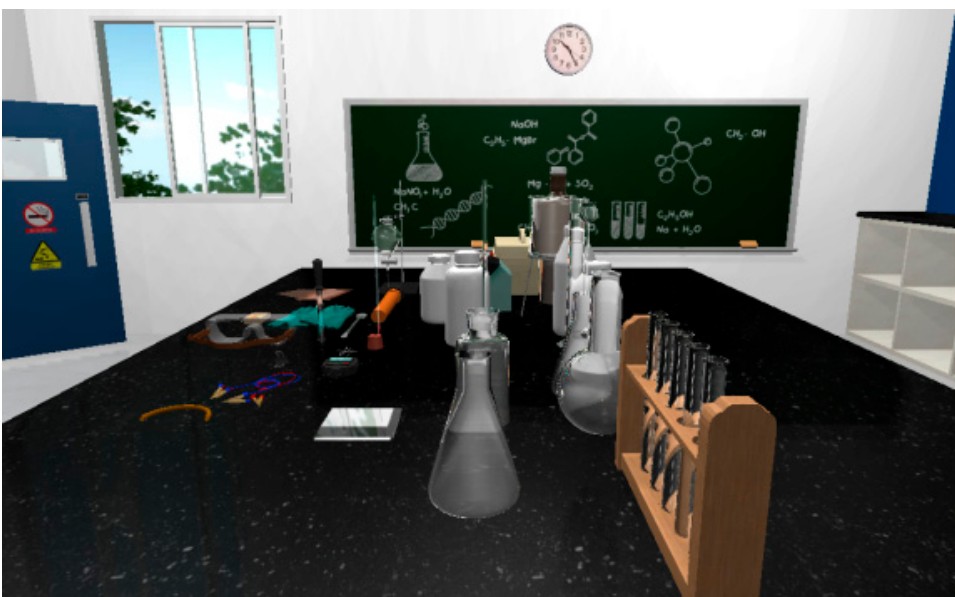

**Figure 5.** Experiment equipment completed collection.

## 5. Research Methods

This study focused on the effect of cognitive loading, and observation of whether experiential learning, cognitive load, and self-efficacy affect learning motivation through the use of a virtual chemical laboratory. This section contains the research model, definition value, questionnaire, participants and procedures, and common method variance.

### 5.1. Research Model

The purpose of this research was to investigate the effects of using a virtual chemistry laboratory on learning motivation and academic achievement. Moreover, this research investigated the moderating effect on learning to understand whether the degree of information involvement will affect academic achievement. The research model is shown in Figure 6.

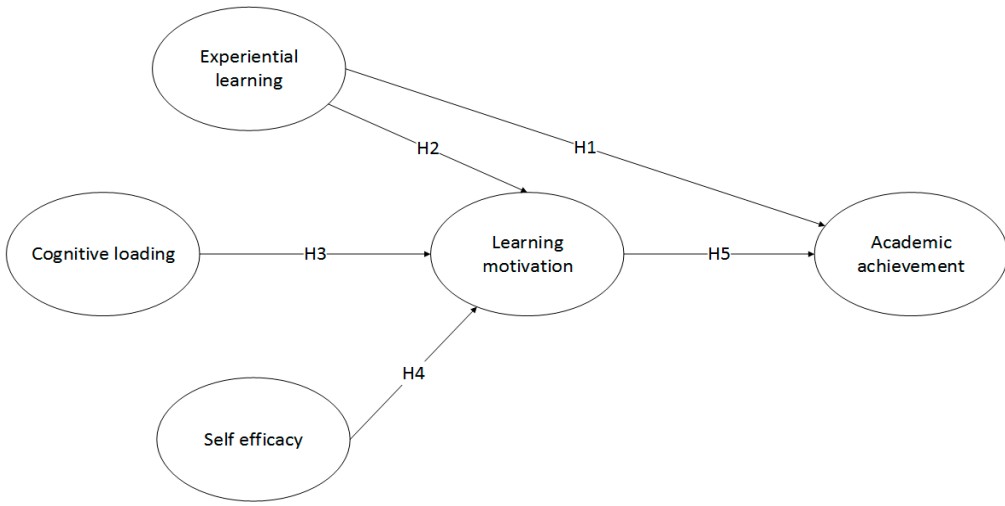

**Figure 6.** Sustainability innovation experiential learning model.

*5.2. Questionnaire*

The questionnaire was divided into two parts. The first part is general information and the second part is the related questions about each dimension. The dimensions in this paper are experiential learning, cognitive load, self-efficacy, learning motivation, and academic achievement. The scale of each dimension in this paper changes according to the situation in which students use the virtual chemistry laboratory. The items in each dimension are of the basic questionnaire design; the questions in each dimension and the scale are shown in Table 3. Responses to all questions are on a 7-point Likert scale, ranging from 7, 'strongly agree', to 1, 'strongly disagree'.

**Table 3.** Items of each dimension.

| Dimension | Item | References |
|---|---|---|
| Experiential learning (EL) | When I study, I don't like the actual operation.<br>I learn though actual operation.<br>When I learn best, it is through implementation and practicing.<br>When I learn, I like to see the results of the implementation.<br>When I am learning, I always try to do something by myself. | [44] |
| Cognitive load (CL) | Compared with other students, I spent a lot of effort on performance in the virtual chemistry laboratory.<br>I cannot concentrate on learning about performance in the virtual chemistry laboratory in the future.<br>It causes me a lot of pressure on the performance in the virtual chemistry laboratory now. | [64] |
| Self-efficacy (SE) | If I do my best, I can always solve the problem.<br>Facing a chemical problem, I usually can find the solution.<br>If I make the necessary efforts, I will be able to solve most of the learning problems.<br>I can face problems calmly because I believe in my problem-solving ability.<br>When I get into trouble, I usually can get the solution. | [65] |

**Table 3.** *Cont.*

| Dimension | Item | References |
|---|---|---|
| Learning motivation (LM) | The beginning of the virtual chemistry laboratory attracts me. For me, the most satisfying thing in the virtual chemistry laboratory is trying my best to understand the course. I am really interested in the content of the virtual chemistry laboratory. When I learn through using the virtual chemistry, I am confident that I can understand the chemical experiment. I enjoy the chemical experiments in the virtual chemistry laboratory, so I want to understand more about chemical experiments. If I can, I want to get better grades than most of the students in this class. | [46,66] |
| Academic achievement (AA) | When I really understand, I know what I have really learned. When I can explain things to others, I know what I have learned. Learning makes me more independent and more confident. I learn a lot through participating in many different contents in the virtual chemistry laboratory. Learning is being taught something I didn't know before. | [67] |

## 5.3. Participants and Procedure

This research focused on investigating how the virtual chemistry laboratory affects learning motivation and academic achievement, thus, the participants were junior high school students with experience in a virtual chemistry laboratory. The questionnaire was divided into two parts; the first part is general information and the second part is the related questions about each dimension. The dimensions in this paper are experiential learning, cognitive load, self-efficacy, learning motivation, and academic achievement. A total of 272 responses were received, and there were no missing data. After collecting the questionnaires, this paper analyzed the data using SPSS and SmartPLS statistical analysis tools. The first part of the questionnaire used descriptive statistical analysis to analyze general information. The second part of the questionnaire used partial least square (PLS). A measurement model was used to carry out confirmatory factor analysis (CFA), and then a structural model was used to investigate the path analysis of the potential factors.

## 6. Data Analysis

This section examines the sustainability innovation experiential learning model and learning effectiveness of the virtual chemistry laboratory in a chemistry course, as analyzed via the pretest and protest of the quasi-experiment. This empirical study is separated into two experimental steps. Step 1: a total of 272 students accessed the survey to evaluate the sustainability innovation experiential learning model; Step 2: the 36 male participants and 36 female participants were randomly selected from 272 responses of tenth grade students. The participants were randomly divided into an experimental group and a control group, and were assigned different approaches with tasks. One group of students served as the control group and the other the experimental group.

## 6.1. Descriptive Statistics

In this study, a total of 272 responses were received and used in analysis. The sample consisted of 145 males (53%) and 127 females (47%). In the aspect of the frequency per week, 2–3 times per week was the most common (52%). In the aspect of the time per frequency, less than 30 minutes was the most common (57%).

## 6.2. Measurement Models and Reliability and Validity Analysis

The results of the reliability analysis are shown in Table 4. To evaluate reliability and validity, this study calculated composite reliability and average variance extracted (AVE) through the PLS algorithm. Cronbach's alpha was used to calculate internal consistency reliability. AVE evaluates construct convergent validity, and the value should be higher than 0.50, which means that the latent construct explains more than half the variance of the indicators. Fornell and Larcker [68] indicated that the criterion of evaluating convergent validity has three points: (1) all outer loading should be higher than 0.50; (2) composite reliability (CR) should be higher than 0.70; and (3) AVE should be higher than 0.50. The outer loading of all constructs are between 0.79–0.85, thus, all of them are higher than the criterion (i.e., 0.50). Next, the CR of each construct is between 0.88–0.91, thus, all of them are higher than the criterion (i.e., 0.70). Finally, the AVE of each construct is between 0.63–0.71, thus, all of them are higher than the criterion (i.e., 0.50). In conclusion, the results indicate that each latent construct has good convergent validity.

Focusing on discriminant validity, Fornell and Larcker [68] suggested that the square root of AVE of the constructs should be higher than other correlations of the constructs, to explain the discriminant validity. Table 4 includes the mean, standard deviation, composite reliability, Cronbach's $\alpha$, and Fornell-Larcker criterion. As shown in Table 4, the square root of AVE in each construct is higher than the correlations of other constructs, which indicates that all constructs have good discriminant validity.

Reliability is defined as the reliability of the measurement data, which means the results derived by the testing tools are consistent. Cronbach's alpha is one of the methods used for evaluation, especially Likert scales. Guilford and Fruchter [69] posit that reliability is high when Cronbach's $\alpha$ is higher than 0.70; otherwise, reliability is low. This research used Cronbach's $\alpha$ and CR to evaluate internal consistency reliability, and the results show that the Cronbach's $\alpha$ of each latent construct is between 0.80–0.88, and the CRs are between 0.88–0.91. The results indicate good consistency between participants and each item of the constructs; also, the results of the survey have good reliability.

**Table 4.** Means Standard Deviations, Reliabilities, and Correlation of Constructs.

| Construct | Mean | STD | Outer Loadings | Cronbach's Alpha | CR | AVE | Fornell-Larcker Criterion | | | | |
|---|---|---|---|---|---|---|---|---|---|---|---|
| | | | | | | | EL | CL | SE | LM | AA |
| EL | 5.93 | 0.96 | 0.81 | 0.87 | 0.91 | 0.66 | 0.81 | | | | |
| CL | 5.72 | 1.23 | 0.85 | 0.8 | 0.88 | 0.71 | −0.58 | 0.84 | | | |
| SE | 5.91 | 0.98 | 0.82 | 0.87 | 0.9 | 0.66 | 0.77 | −0.54 | 0.81 | | |
| LM | 5.97 | 0.95 | 0.83 | 0.88 | 0.91 | 0.68 | 0.73 | −0.58 | 0.72 | 0.82 | |
| AA | 5.29 | 1.06 | 0.79 | 0.85 | 0.91 | 0.63 | 0.7 | −0.6 | 0.66 | 0.81 | 0.8 |

Experiential learning (EL), Cognitive load (CL), Self-efficacy (SE), Learning motivation (LM), Academic achievement (AA).

## 6.3. Evaluation of the Structural Model

This research used PLS (SmartPLS) to test each hypothesis. After confirming that each construct has good reliability and validity, this research evaluated the structural model, including evaluating the prediction of the model and the relationship between the constructs. The evaluation of the model has five steps: (1) Evaluate the collinearity; (2) Evaluate the significance and the correlation of the structural model; (3) Evaluate the coefficient of determination ($R^2$); (4) Evaluate the $f^2$ effect size; and (5) Evaluate the predictive relevance ($Q^2$) [70].

## 6.4. Evaluation of the Collinearity

This research used the variance inflation factor (VIF) to evaluate the collinearity problem. When the VIF of the predicted construct is higher than 5.00, the predicted construct has a collinearity problem [70]. Table 5 shows that the VIF of all the variables is lower than 5.00, thus, this research does not have a collinearity problem.

**Table 5.** Factor Loadings and Cross Loadings for the Measurement Model.

| Items | EL | CL | SE | LM | AA | VIF |
|-------|------|-------|------|-------|-------|------|
| EL_1 | 0.82 | −0.48 | 0.71 | 0.65 | 0.63 | 2.14 |
| EL_2 | 0.84 | −0.49 | 0.60 | 0.54 | 0.54 | 2.28 |
| EL_3 | 0.84 | −0.43 | 0.70 | 0.68 | 0.61 | 2.21 |
| EL_4 | 0.81 | −0.47 | 0.55 | 0.56 | 0.57 | 2.07 |
| EL_5 | 0.76 | −0.49 | 0.52 | 0.49 | 0.47 | 2.04 |
| CL_1 | −0.52 | 0.85 | −0.45 | −0.49 | −0.48 | 1.82 |
| CL_2 | −0.47 | 0.86 | −0.47 | −0.48 | −0.51 | 1.93 |
| CL_3 | −0.48 | 0.82 | −0.45 | −0.49 | −0.54 | 1.54 |
| SE_1 | 0.54 | −0.37 | 0.82 | 0.50 | 0.49 | 2.33 |
| SE_2 | 0.61 | −0.42 | 0.78 | 0.60 | 0.55 | 1.71 |
| SE_3 | 0.66 | −0.47 | 0.82 | 0.59 | 0.52 | 2.22 |
| SE_4 | 0.64 | −0.44 | 0.82 | 0.58 | 0.56 | 2.04 |
| SE_5 | 0.64 | −0.47 | 0.82 | 0.61 | 0.54 | 1.99 |
| LM_1 | 0.59 | −0.49 | 0.62 | 0.81 | 0.67 | 2.02 |
| LM_2 | 0.53 | −0.41 | 0.53 | 0.81 | 0.62 | 2.34 |
| LM_3 | 0.61 | −0.52 | 0.62 | 0.87 | 0.69 | 2.62 |
| LM_4 | 0.67 | −0.55 | 0.61 | 0.84 | 0.68 | 2.30 |
| LM_5 | 0.58 | −0.42 | 0.56 | 0.79 | 0.69 | 2.03 |
| AA_1 | 0.62 | −0.54 | 0.66 | 0.70 | 0.80 | 1.78 |
| AA_2 | 0.59 | −0.50 | 0.52 | 0.69 | 0.86 | 2.49 |
| AA_3 | 0.58 | −0.50 | 0.52 | 0.69 | 0.86 | 2.50 |
| AA_4 | 0.48 | −0.44 | 0.45 | 0.89 | 0.74 | 1.58 |
| AA_5 | 0.50 | −0.40 | 0.46 | 0.55 | 0.72 | 1.57 |

## 6.5. Path Coefficient of the Structural Model

After the use of the PLS algorithm, the path coefficient of the structural model was conducted. The standardized values of the path coefficients should be between −1 and +1. The value that is positive and highly correlated is closer to +1; conversely, the value that is negative and highly correlated is closer to −1. The closer the value to 0, the weaker the relationship. The significance of the path coefficient is determined by the empirical t value through bootstrapping [70]. Due to the sample being more than 30, the data are close to t-distribution. This research used the quartile of normal distribution as the critical value to compare the empirical t values of the data.

As shown in Figure 7, the results indicate that the path coefficient of H1 is 0.231 and the empirical t value is 3.728. Due to the value being higher than the t value, which is 1.96 when the significant level is 5%, H1 is significant and positively correlated. The path coefficient of H2 is 0.357 and the empirical t value is 4.762. H2 is significant and positively correlated. The path coefficient of H3 is −0.19 and the empirical t value is 3.02. H3 is significant and negatively correlated. The path coefficient of H4 is 0.339 and the empirical t value is 4.159. H4 is significant and positively correlated. The path coefficient of H5 is 0.644 and the empirical t value is 11.403. H5 is significant and positively correlation.

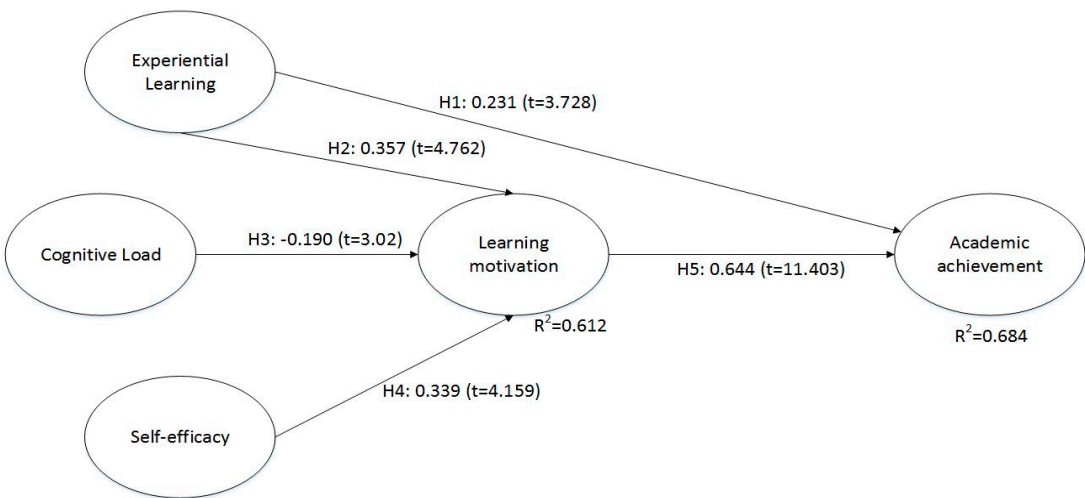

**Figure 7.** Path coefficient of the structural model.

### 6.5.1. Coefficient of Determination ($R^2$)

The coefficient of determination is used to measure the accuracy of the prediction regarding the overall effectiveness between all the exogenous variables and endogenous variables. The value of $R^2$ is between 0 and 1; a higher value represents higher prediction. $R^2$ values of 0.75, 0.50, or 0.25 are described as substantial, moderate, and weak, respectively [70]. In the path coefficient of the model (Figure 7), $R^2$ of learning motivation (LM) and academic achievement (AA) are described as moderate ($R^2 = 0.612$, $R^2 = 0.684$).

### 6.5.2. $f^2$ Effect Size

The $f^2$ effect size can be used to evaluate whether or not the deleted exogenous construct has significant effectiveness, and the calculation was based on Equation (1). The difference is between including and excluding the exogenous construct. The principal of evaluating $f^2$ was classified as 0.02, 0.15, and 0.35, which is described as the small, medium, and large effect, respectively [71]. In the path coefficient of the model (Figure 7), the $R^2_{included}$ value of LM is 0.612; relatively, the $R^2_{excluded,EL}$ value is 0.565, the $R^2_{excluded,CL}$ value is 0.589, and the $R^2_{excluded, SE}$ value is 0.567. Therefore, the $f^2$ effect size, of which the endogenous constructs experiential learning (EL), cognitive load (CL) and eelf-efficacy (SE) explain the exogenous variable LM, are 0.121, 0.059, and 0.115, respectively. The $f^2$ effect size between EL and LM is medium and the $f^2$ effect sizes among CL, SE, and LM are both small. The $R^2_{included}$ value of AA is 0.684; relatively, the $R^2_{excluded,EL}$ value is 0.659, and the $R^2_{excluded, LM}$ value is 0.489. Therefore, the $f^2$ effect size of which the endogenous constructs EL and LM explain the exogenous variable AA are 0.079 and 0.61, respectively. The $f^2$ effect size between EL and AA is small and the $f^2$ effect size between EL and AA is large:

$$f^2 = \frac{R^2_{included} - R^2_{excluded}}{1 - R^2_{included}} \tag{1}$$

### 6.5.3. Predictive Relevance ($Q^2$)

$Q^2$ is one of indices of predictive relevance. When the results are confirmed as predictive relevance, it means that the model can accurately predict the endogenous construct of the reflective measurement model and the indices of the endogenous construct in one item. In the structural model, the $Q^2$ value of the reflective endogenous latent construct was higher than 0, which represents that the path models and the construct have predictive relevance. This research used cross- validated redundancy to calculate the $Q^2$ value, as the method used the key element of the path models and the structural model to predict the deleted data. The results indicate that the $Q^2$ value of LM is 0.416 and the $Q^2$

value of AA is 0.43. Both of the $Q^2$ values are higher than 0, which indicates that both have predictive relevance. In addition, the relative importance can be compared through the $q^2$ value, and the function is shown in Equation (2). The results represent that the $Q^2_{included}$ value of LM, which includes EL, CL, and SE, is 0.416. The $Q^2_{excluded}$ value of LM, which excluded EL, is 0.38. The $Q^2_{excluded}$ value of LM, which excluded CL, is 0.40. The $Q^2_{excluded}$ value of LM, which excluded SE, is 0.384. Therefore, the $q^2$ values that EL, CL, and SE explaining the endogenous construct LM are 0.061, 0.027, and 0.054, respectively. All of the $q^2$ values have small predictive relevance. The $Q^2_{included}$ value of AA, which included EL and LM, is 0.43. The $Q^2_{excluded}$ value of AA, which excluded EL, is 0.414. The $Q^2_{excluded}$ value of AA, which excluded LM, is 0.306. Therefore, the $q^2$ values that EL and LM explained the endogenous construct AA are 0.028 and 0.217, respectively. The $q^2$ value that EL explained to AA has small predictive relevance. The $q^2$ value that LM explained to AA has medium predictive relevance:

$$q^2 = \frac{Q^2_{included} - Q^2_{excluded}}{1 - Q^2_{included}} \tag{2}$$

### 6.5.4. Global Goodness of Fit (GoF)

This research used the GoF, as proposed by Tenenhaus, Vinzi, Chatelin and Lauro [72], to confirm the global validity of the model, and the function is shown in Equation (3). The results indicate that the GoF is 0.658, which is higher than the value of $GoF_{large}$ ($GoF_{small} = 0.1$, $GoF_{medium} = 0.25$, $GoF_{large} = 0.36$). Therefore, the model in this research has good explanation and fully validates the PLS model [73]. Table 6 shows the values of the structural model:

$$GoF = \sqrt{\overline{AVE} \times \overline{R^2}} \tag{3}$$

**Table 6.** The values of the structural model.

| Construct | $R^2$ | $f^2$ (LM) | $f^2$ (AA) | $q^2$ (LM) | $q^2$ (AA) |
|-----------|-------|------------|------------|------------|------------|
| EL | | 0.121 | 0.079 | 0.061 | |
| CL | | 0.059 | | 0.027 | |
| SE | | 0.115 | | 0.054 | |
| LM | 0.612 | | 0.489 | | 0.028 |
| AA | 0.684 | | | | 0.217 |

### 6.5.5. Important-Performance Matrix Analysis

One of the PLS features of the extracted score of the latent construct is important-performance matrix analysis (IPMA), which is the score used to extend the original result. Basic PLS analysis is used to confirm the relative importance of the construct through the evaluation of all related paths in the structural model. IPMA also considers the performance of the construct. The goal construct should be set when executing IPMA. Before completing the goal construct, the global effect (i.e., performance) should be obtained. The degree of importance that the latent variables affect the endogenous construct is decided according to the global effect of these variables [74]. Through adjusting the latent variable, the values were rescaled to 0–100, as shown in Equation (4). $\xi_i$ represents the exogenous variable i in the inner model. $E[.]$, min[.], and max[.] represent the expected minimum and maximum. In Equations (5) and (6), the minimum and maximum were dependent on the variables of the related measurement model [75]. $w_{ij}$ represents the $i$th value of the $j$th latent variable. Each average of the recalled latent variable could define the performance, where the higher value represents the better performance.

$$\xi_i^{rescaled} = \frac{E[\xi_i] - \min[\xi_i]}{\max[\xi_i] - \min[\xi_i]} \cdot 100 \tag{4}$$

$$\min[\xi_i] = \sum_{j=1}^{n_i} w_{ij} \cdot \min\left[x_{ij}\right] \tag{5}$$

$$\max[\xi_i] = \sum_{j=1}^{n_i} w_{ij} \cdot \max\left[x_{ij}\right] \tag{6}$$

Table 7 shows the importance and performance in IPMA. The importance of LM is 0.681, which indicates the most important construct, while performance is 75.871, which is the worst. EL is second most important, and its performance is the best, which is 79.766. The average importance and performance of the four latent construct are drawn in a two-dimensional matrix and divided into four quadrants; keep up, do better, education, and no change (see Figure 8), to evaluate the importance and the performance of the construct. EL and LM are located in the keep up area, as their importance and performance are both high, which indicates that experiential learning and learning motivation have important and significant effect on academic achievement. SE is located in the education area, and its importance is low, while its performance is high. Although students perform well in self-efficacy, the mportance of self-efficacy is low, thus students may focus on self-efficacy excessively. CL is located in the no change area, as its importance and performance are low, which indicates that cognitive load does not need to improve, and would have the least positive effect.

**Table 7.** Important-performance matrix analysis (IPMA).

|  | AA | | Important | Performance |
|---|---|---|---|---|
|  | Direct | Indirect | Total Effect |  |
| EL | 0.23 | 0.23 | 0.43 | 82.46 |
| CL |  | −0.12 | −0.12 | 21.15 |
| SE |  | 0.22 | 0.22 | 82.12 |
| LM | 0.64 |  | 0.64 | 83.04 |

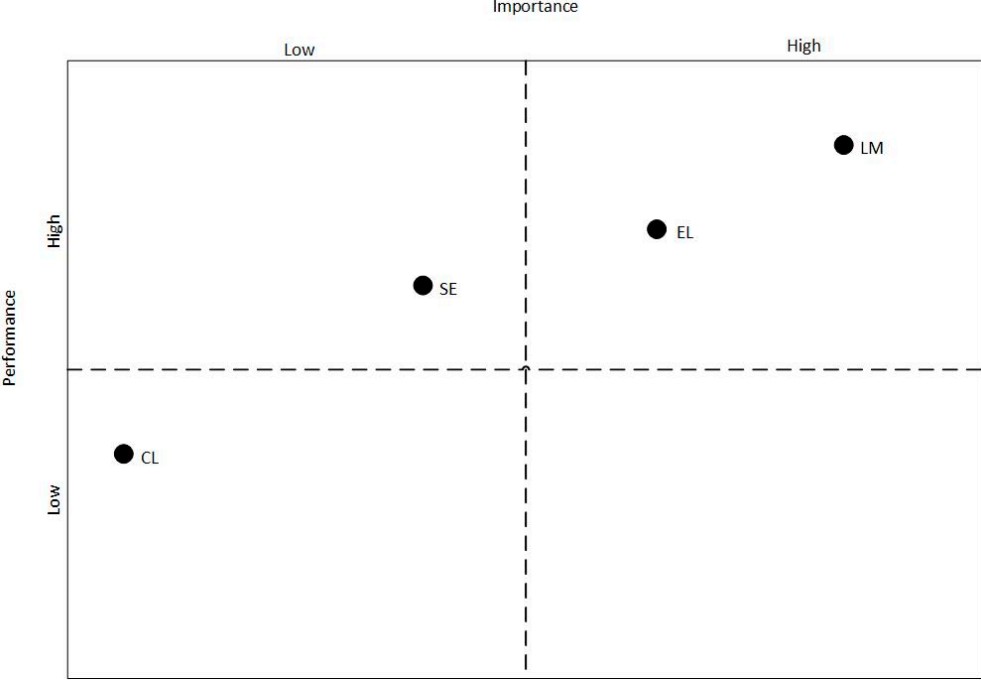

**Figure 8.** IPMA matrix for Sustainability Innovation Experiential Learning Model.

### 6.5.6. The Validation of Students' Academic Achievement

This study found that, according to survey analysis, experiential learning and learning motivation positively affect academic achievement. To investigate the difference of grades between before and after using the chemistry laboratory system, this research investigated the difference between the pretest and posttest through quasi experimentation. The 36 male participants and 36 female participants were random selected from 272 responses from tenth grade students. The participants were randomly divided into an experimental group and a control group, and were assigned different approaches with tasks. One group of students served as the control group and the other the experimental group. The period of the experimental activity was nine weeks, with four phases. In the first phase (the first week), both groups were introduced to the fundamental chemical concepts. In the second phase (the second week), all of the students took the first exam and their grades were collected. In the third phase (the third to eighth week), the participants of the control group used the traditional course, while the participants of the experimental group used the virtual chemical laboratory to assist learning the chemical concepts. Finally, in the fourth phase (the ninth week), both groups took the second exam and the grades were collected (as Figure 9).

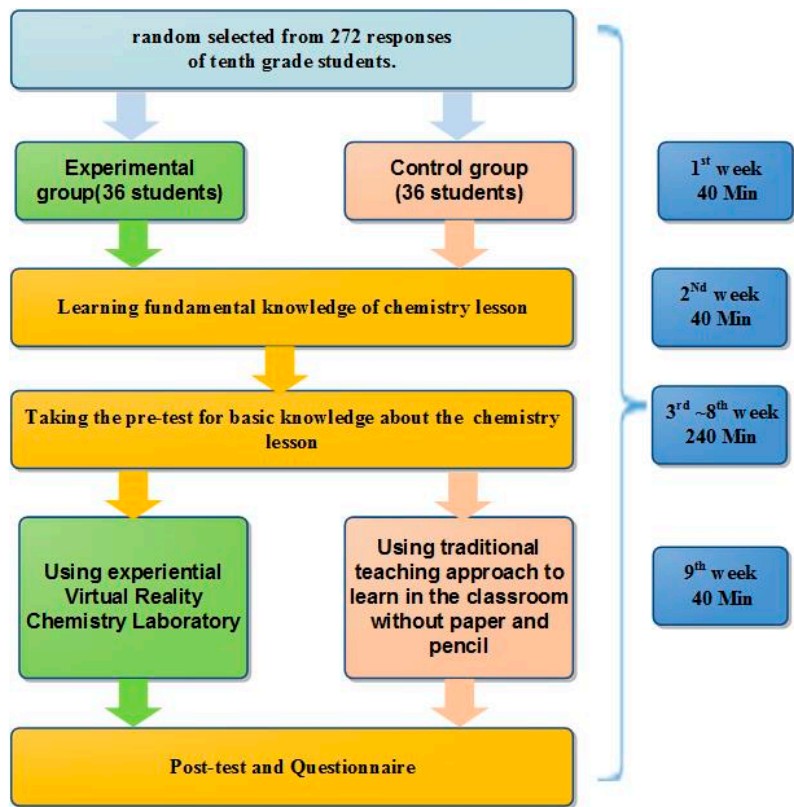

**Figure 9.** Sustainability innovation experiential learning experimental process.

To avoid repeated achievement testing, the grades of the first chemical exam were used as the covariance. This research used one-way ANOVA to investigate whether using the virtual chemistry laboratory system affected the students' academic achievement. The differences of the pre- and post-test achievements of the experimental group and control group are shown in Table 8. To investigate whether or not there is a significant difference between the traditional course and the virtual chemical laboratory, this study used Levene's test to examine the pre- and post-test grades before performing one-way ANOVA. The result of Levene's test for equality of variances indicates that the assumption of the homogeneity of variances in the group is satisfied ($p > 0.05$). The result of ANOVA shows that there was no significant difference between the two groups on pre-test (F = 0.635, $p = 0.428 > 0.05$). Therefore,

it is evident that the students in the two groups had equivalent prior knowledge before the learning activity. According to the results, there are significant differences in the post-test after the learning activity (F = 26.086, $p$ = 0.000 < 0.05). Thus, the grades of the experimental group who used the virtual chemical laboratory to support leaning were better than the control group. Figure 10 presents the results of the exam, which clearly exhibit positive and significant influences. In the pre-test, there is no significant difference in the grades between the control group (M = 76.34, SD = 7.52) and the experimental group (M = 77.71, SD = 6.86). However, there is significant difference in the post-test between the experimental group (M = −87.17, SD = 7.47) and control group (M = 77.82, SD = 7.82). Thus, the experimental group had better improvement and the average score is better when using the virtual chemistry laboratory system.

**Table 8.** The difference of pre-and post-test achievement of different groups.

|  | Group | N | Mean | SD | SE | F | Sig. |
|---|---|---|---|---|---|---|---|
| pretest | Control | 36 | 76.34 | 7.52 | 1.27 | 0.635 | 0.428 |
|  | Experimental | 36 | 77.71 | 6.86 | 1.16 |  |  |
| posttest | Control | 36 | 77.82 | 7.82 | 1.32 | 26.086 | 0.000 |
|  | Experimental | 36 | 87.17 | 7.47 | 1.26 |  |  |

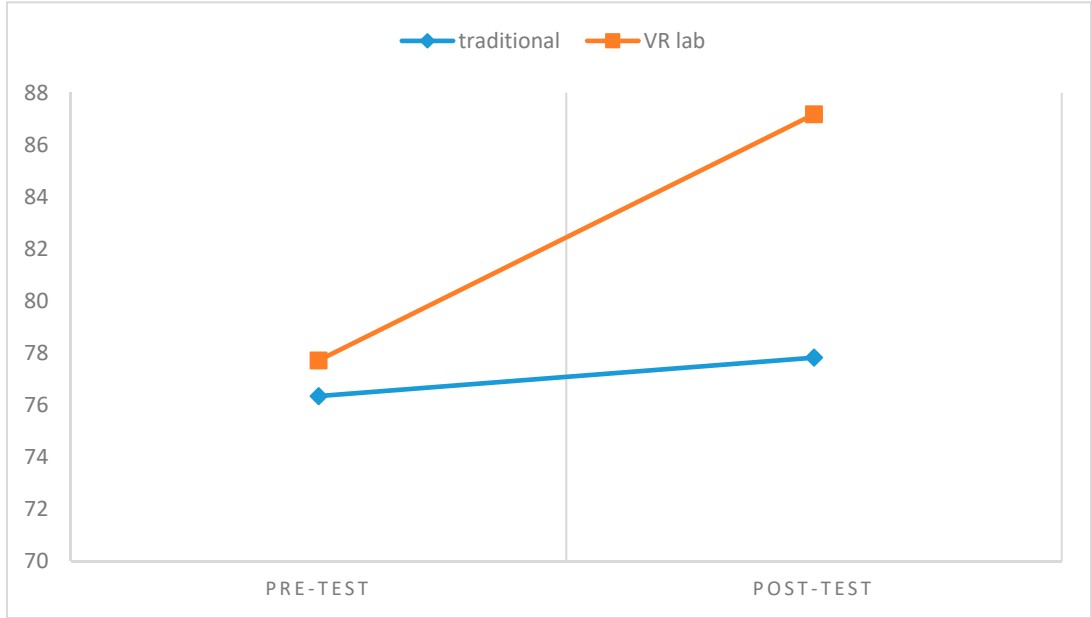

**Figure 10.** Mean score for pre- and post-test achievement of experimental group and control group.

## 7. Discussion

The main objective of this study was to investigate the influence of the virtual chemistry laboratory on academic achievement. This study investigated the relationships among students' self-efficacy, cognitive load, and learning motivation, as well as the relationship between students' learning motivation and academic achievement, though experiential learning using a virtual chemistry laboratory. The results of this study found that experiential learning improved students' learning motivation. The experiential results indicate that the students who used the virtual chemistry laboratory system had higher grades than their pretest. This confirms that using the virtual chemistry laboratory system could improve students' academic achievements, and explained that there is a positive relationship between learning motivation and academic achievement. According to the literature review and empirical study, this study summarized the theoretical and practical implications, and discussed the limitations and future work. This study aimed to investigate how a virtual

chemistry laboratory influences academic achievement. There are four findings, which are summarized, as follows.

*7.1. The Relationships among Experiential Learning, Learning Motivation, and Academic Achievement*

According to the results, experiential learning affects academic achievement significantly in the structural model of the virtual chemistry laboratory, which indicates that the students who participated in the experiential learning had good academic achievement. Students can transform their experiences into knowledge through the game experience to improve their academic achievement. Previous studies also indicated that students obtain practical experience when experiential learning is applied to teaching and learning, as it engages students to learn skills and makes learning more effective [48,49].

The results indicate that experiential learning affects learning motivation significantly in the structural model of the virtual chemistry laboratory, which indicates that the students who participated in the experiential learning had high learning motivation. Constructivists believe that an individual's learning is obtained from experiences, and the process of knowledge construction happens when learners are engaged in their own meaningful tasks [47]. Therefore, experiential learning strongly engages student's learning motivation.

*7.2. The Relationship between Cognitive and Motivation*

In the structural model of the virtual chemistry laboratory, cognitive load negatively affects learning motivation, which indicates that students need more memory in learning courses. The results show that students can obtain learning motivation through the design of the virtual chemistry laboratory and achieve good learning performance. When students' cognitive load is high and overloaded, students cannot engage in critical thinking or completely understand the proffered knowledge [76].

*7.3. The Relationship between Self-Efficacy and Learning Motivation*

In the structural model of the virtual chemistry laboratory, self-efficacy positively affects learning motivation, which indicates that students with high self-efficacy have strong learning motivation. Self-efficacy has a positive relationship between intrinsic motivation and academic achievement [60]. In other words, those with higher self-efficacy can acquire intrinsic motivation and improve their learning performance. An individual's ability depends on optimistic or pessimistic thinking, thus, when low self-efficacy students face difficulties, they give up easily [61].

*7.4. The Relationship between Learning Motivation and Academic Achievement*

In the structural model of the virtual chemistry laboratory, learning motivation positively affects academic achievement, which indicates that when students' motivation is stronger, they have better academic achievement. As previous studies pointed out, learning motivation has a significant relationship with academic achievement. Learning motivation is helpful and constructive to academic achievement [77,78]. The higher the students' motivation, the higher their academic achievements. The results of the five hypotheses were all supported.

## 8. Conclusions

*8.1. Theoretical Implications*

As the technical development of VR has grown, many studies have been conducted in the field of technology education investigating students' academic achievement. However, few have investigated students' self-efficacy and cognitive load to verify whether they influence students' learning motivation through using the VR system. Therefore, this study investigated whether or not using a virtual reality system improves students' cognition and self-efficacy and cognitive load on learning chemical concepts. Moreover, this study considered whether the VR system influenced students' academic

achievement. In addition to experiential learning, the research model added self-efficacy and cognitive load. This research empirically analyzed the effects on students' learning motivation and academic achievement through experiential learning and the virtual chemistry laboratory, in order to analyze whether or not students' motivation was due to the improvement of self-efficacy. The results found positive relationships between self-efficacy and learning motivation, and between cognitive load and learning motivation, which represents that experiential learning helps students transform their practical experience into knowledge. In addition, the improvement of students' learning motivation positively affects students' academic achievement.

### 8.2. Practical Implications

In the aspect of industry, the results indicate that using VR experiential learning is an important factor and can improve students' learning motivation. Thus, this research suggests that industry could consider its application in education to prompt education pluralism when designing VR products.

In the educational aspect, chemical experiments are dangerous and costly. The results suggest that, before conducting real experiments, students become familiar with the experimental process by training in the virtual chemistry laboratory, thereby reducing the possibility of accidents. Moreover, students can use the laboratory in their spare time, which supports students' learning process. The contribution of the study is to propose a sustainable innovation learning model and use the IPMA to analyze the important design concept for the virtual reality chemistry laboratory simulation game. This may pose different learning outcomes, because simulations and games have different design features and interactive models. The research result could provide design guidelines for experiential virtual reality learning materials.

### 8.3. Limitations and Future Work

This research has some limitations and suggestions to improve future work. As the period of the experimental activity was short, it might be difficult to claim that all of the findings are significant. It is possible that the results would partially change when the period of the experimental activity is longer, meaning students' interest and curiosity may be reduced; thus this research suggests that the period of experimental activity could be longer. Future work could evaluate the VR educational game for students of different ages in different schools to investigate more differences. This research focused on improving the students' effectiveness in learning chemistry concepts; future work could focus on different extracurricular contents and support better learning effectiveness through more games or tasks.

**Author Contributions:** Conceptualization, C.-H.S.; Methodology, C.-H.S.; Software, C.-H.S.; Validation, C.-H.S.; Formal Analysis, C.-H.S. and T.-W.C.; Investigation, T.-W.C.; Writing-Original Draft Preparation, C.-H.S. and T.-W.C.; Writing-Review & Editing, C.-H.S.; Visualization, C.-H.S.; Supervision, C.-H.S.; Project Administration, C.-H.S.

**Funding:** This research was funded by the Ministry of Science and Technology, Taiwan grant number MOST 107-2637-H-366-002-.

**Acknowledgments:** The authors would like to thanks the Ministry of Science and Technology, Taiwan.

**Conflicts of Interest:** The author declares no conflict of interests.

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
