# Peer review of "A Sustainability Innovation Experiential Learning Model for Virtual Reality Chemistry Laboratory: An Empirical Study with PLS-SEM and IPMA"

_sustainability, doi:10.3390/su11041027_

Reviewer 1 Report

Manuscript ID: sustainability-435443

The paper “A Sustainability Innovation Experiential Learning Model for 3D VR Chemistry Laboratory: An Empirical Study with PLS-SEM and IPMA” 

This paper focuses on the effect of cognitive loading and observes whether the flow, cognitive load, and self-efficacy affect students’ learning motivation, through the use of a VR chemical laboratory.

First of all, my opinion is that this paper is too long and difficult to read, and the structure of the paper may be improved.

Title of the paper: I suggest to eliminate 3D from the title VR means Virtual Reality that it obviously a 3D environment.

Keywords: I suggest to the author to check the keywords. I think is better to use also VR- environment, or Chemical VR Lab, serious games so please substitute some of the present keywords with the suggested ones.

Check the Sustainable Innovation Learning, I think you miss some capital letters.

Introduction section: I suggest to improve this section using this paragraph to introduce briefly the research context (Sustainable Innovation Learning) and to declare the objectives of the work and its outline (see lines 90-95).

Typos: Check HEI or HEI as the abbreviation of Higher Educational Institutes, I think the correct form is Higher Education Institution - HEIs

The part dealing with VR and chemistry must be moved in the next section.

Section 2 Related Work: I suggest to change the name of this section in Background because here the author explains some concept need to understand the context of his work.

I suggest to rename Subsection 2.3 as a new paragraph entitled “Research Hypothesis” and to move here the description and explanation of your research model (fig 6).

Section 3 Material Design and System

I suggest renaming this section as “4 The VR Chemistry laboratory framework”

Figures: I suggest to use English for all the text present in the figures – see fig 4

Section 4 Method: rename as: “5 Research Methods”

I suggest to explain better the participant sample, introduce a table, and the pre and post-test procedures.

The author talks about 272 students then of two groups of 36 students while in table 8 he uses the number 35?

Section 5: rename as “ 6 Data Analysis”

If it is possible to reduce the general statistical consideration of this section highlighting the specific statistical consideration obtained by the analysis of the data sample.

Section 6 Discussion and Conclusion.

I suggest to split this section into two sections: “7 Discussion”  from line 544 to 594  and “8 Conclusions” from line 595.

The English language must be improved see for example line 41-44.

Author Response

Cover Letter

Dear Editor

This is Manuscript ID sustainability-435443 major revisions of Special Issue "Selected Papers from 2nd Eurasian Conference on Educational Innovation 2019". The article title: “A Sustainability Innovation Experiential Learning Model for 3D VR Chemistry Laboratory: An Empirical Study with PLS-SEM and IPMA”.

The revisions of this paper have been clearly highlighted using the "Track Changes", and this paper has been edited by a native English speaker.

This cover letter detailing any changes as below:

Reviewer 1:

This paper focuses on the effect of cognitive loading and observes whether the flow, cognitive load, and self-efficacy affect students’ learning motivation, through the use of a VR chemical laboratory.

Question 1: First of all, my opinion is that this paper is too long and difficult to read, and the structure of the paper may be improved.

ANS: Thank you for your suggestion and author modify it. You can see all the "Track Changes" on all the sections.

Question 2: Title of the paper: I suggest to eliminate 3D from the title VR means Virtual Reality that it obviously a 3D environment.

ANS: Thank you for reviewer’s comment, Thank you for reviews kindly suggestions and author have modified it. The new title of the paper:” A Sustainability Innovation Experiential Learning Model for Virtual Reality Chemistry Laboratory: An Empirical Study with PLS-SEM and IPMA”.

Question 3: Keywords: I suggest to the author to check the keywords. I think is better to use also VR- environment, or Chemical VR Lab, serious games so please substitute some of the present keywords with the suggested ones.

ANS: Thank you for reviews kindly suggestions and author have modified it.

Question 4: Check the Sustainable Innovation Learning, I think you miss some capital letters.

ANS: Thank you for your suggestion and author modify it. Author change it to “Sustainable Innovation Learning Model”

Question 5: Introduction section: I suggest to improve this section using this paragraph to introduce briefly the research context (Sustainable Innovation Learning) and to declare the objectives of the work and its outline (see lines 90-95).

ANS: Thank you for your suggestion and author modify it. Authors have rewrite it (see lines 28-95)

Question 6: Typos: Check HEI or HEI as the abbreviation of Higher Educational Institutes, I think the correct form is Higher Education Institution – HEIs

ANS: Thank you for your suggestion and author modify it.

Question 7: The part dealing with VR and chemistry must be moved in the next section.

ANS: Thank you for your suggestion and author modify it.

Question 8: Section 2 Related Work: I suggest to change the name of this section in Background because here the author explains some concept need to understand the context of his work.

ANS: Thank you for your suggestion and author modify it.

Question 9: I suggest to rename Subsection 2.3 as a new paragraph entitled “Research Hypothesis” and to move here the description and explanation of your research model (fig 6).

ANS: Thank you for your suggestion and author modify it.

Question 10 : Section 3 Material Design and System I suggest renaming this section as “4 The VR Chemistry laboratory framework”.

ANS: Thank you for your suggestion and author modify it.

Question 11: Figures: I suggest to use English for all the text present in the figures – see fig 4

ANS: Thank you for your suggestion and author modify it. see fig 4

Question 12: Section 4 Method: rename as: “5 Research Methods”

ANS: Thank you for your suggestion and author modify it.

Question 13: I suggest to explain better the participant sample, introduce a table, and the pre and post-test procedures.

ANS: Thank you for your good and valuable suggestion but in this study, there are many statistical data table to explain the result. By the way, author have been present the 6.5.6. The validation of students’ academic achievement for the pre and post-test procedures.

Question 14: The author talks about 272 students then of two groups of 36 students while in table 8 he uses the number 35?

ANS: Thank you for your suggestion and author modify it. See line 360~367

This section examines the sustainability innovation experiential learning model and learning effectiveness of the virtual chemistry laboratory in a chemistry course, as analyzed via the pretest and protest of the quasi-experiment. This empirical study are separate into two steps experiment. Step 1: a total of 272 students accessed the survey to evaluate the sustainability innovation experiential learning model; Step 2: the 36 male participants and 36 female participants were random selected from 272 responses of tenth grade students. The participants were randomly divided into an experimental group and a control group, and were assigned different approaches with tasks. One group of students served as the control group and the other the experimental group.

Question 15: Section 5: rename as “ 6 Data Analysis”

ANS: Thank you for your suggestion and author modify it.

Question 16: If it is possible to reduce the general statistical consideration of this section highlighting the specific statistical consideration obtained by the analysis of the data sample.

ANS: Thank you very much for reviewer’s suggestions, and author modify it. See table 6 and Table 7 by the analysis of the data sample.

Question 17: Section 6 Discussion and Conclusion. I suggest to split this section into two sections: “7 Discussion”  from line 544 to 594  and “8 Conclusions” from line 595.

ANS: Thank you for your suggestion and author modify it.

Question 19: The English language must be improved see for example line 41-44.

ANS: Thank you for your suggestion and author modify it. You can see the tracking change in red color words.

Reviewer 2 Report

Authors in this paper us on serious virtual reality games and investigate whether or not a virtual reality system improves students performance on chemical concepts.

The work is interesting and in general it is well written. However, it needs extensions and improvements across all of its aspects.

The introduction needs to present more explicitly the exact purpose of the study and the research objectives that author aims to address.  Please highlight the importance and the contribution of the study.

The related work section needs to present additional recent related works mainly on the topics of science (in general). Specifically, author could examine works on the topics of the use of virtual reality environments and 3D virtual worlds in generic science topics.

In this regards, I suggest author to create a new sub-section, name it virtual reality in science and examine related recent works

Also, the existing related works need to be better presented and examined. Author could extend each one and highlight better the main findings of each one. Please highlight the importance and the contribution of the hypotheses

The characteristics of the virtual reality chemical laboratory need to be resented in better detail.

Please explain the learning capabilities of the environment.

The method as presented in section 4 needs a better presentation and its rational needs an explanation.

The results are interesting. I feel that a better discussion is needed and a better comparison with related works in the literature.

Please explain how the results and the findings of the study will be assistive the related research community.

Author Response

Cover Letter

Dear Editor

This is Manuscript ID sustainability-435443 major revisions of Special Issue "Selected Papers from 2nd Eurasian Conference on Educational Innovation 2019". The article title: “A Sustainability Innovation Experiential Learning Model for 3D VR Chemistry Laboratory: An Empirical Study with PLS-SEM and IPMA”.

The revisions of this paper have been clearly highlighted using the "Track Changes", and this paper has been edited by a native English speaker.

This cover letter detailing any changes as below:

Reviewer 2:

Question 1: The work is interesting and in general it is well written. However, it needs extensions and improvements across all of its aspects.

ANS: Thank you for reviewer's appreciate. Your comments will encourage me to write more quality paper.

Question 2: The introduction needs to present more explicitly the exact purpose of the study and the research objectives that author aims to address.  Please highlight the importance and the contribution of the study.

ANS: Thank you for your suggestion and author modify it. See line 80 to line 95 red color words.

Question 3: The related work section needs to present additional recent related works mainly on the topics of science (in general). Specifically, author could examine works on the topics of the use of virtual reality environments and 3D virtual worlds in generic science topics.

ANS: Thank you for your suggestion and author modify it. Authors have added recently reference in 2015~2019 on the topics of the use of virtual reality environments and 3D virtual worlds.

(Simon, S. C., & Greitemeyer, T.,2019)

(Kim, D., & Ko, Y. J.,2019)

(Marou sek, 2013; Mardoyan and Braun, 2015; Bieber et al., 2018; Fazey et al., 2018).

(Van , Brengman, & Willems,2017)

(Hammick & Lee, 2014; Didehbani,Allen,Kandalaft,Krawczyk,& Chapman,2016).

(Berenice Serrano, Rosa M. Ba~nos, Cristina Botella,2016)

(Peperkorn, Diemer & Mühlberger, 2015).

(Ángel, 2015).

Question 4: In this regards, I suggest author to create a new sub-section, name it virtual reality in science and examine related recent works.

ANS: Thank you for your suggestion and author modify it. You can see the tracking change in red color.

Question 5: Also, the existing related works need to be better presented and examined. Author could extend each one and highlight better the main findings of each one. Please highlight the importance and the contribution of the hypotheses.

ANS: Thank you for your suggestion and author modify it. See table 1.

Question 6: The characteristics of the virtual reality chemical laboratory need to be presented in better detail. Please explain the learning capabilities of the environment.

ANS: Thank you for your suggestion and author modify it. See line 134~162.

Question 7: The method as presented in section 4 needs a better presentation and its rational needs an explanation.

ANS: Thank you for your suggestion and author modify it. Based on the other reviewer kindly suggestion to change section 4 Method: rename as: “5 Research Methods”. Authors have rename and better explanation on it. See line 326~359.

Question 8: The results are interesting. I feel that a better discussion is needed and a better comparison with related works in the literature.

ANS: Thank you for your suggestion and author modify it. See line 558 to line 607.

Some of better comparison with related works in the literature. Previous studies also indicated that students obtain practical experience when experiential learning is applied to teaching and learning, as it engages students to learn skills and makes learning more effective (Shokoohi et al., 2016; Wu et al., 2016).When students’ cognitive load is high and overloaded, students cannot engage in critical thinking or completely understand the proffered knowledge (Josephsen, 2015). Self-efficacy has a positive relationship between intrinsic motivation and academic achievement (Hsia et al., 2016). When low self-efficacy students face difficulties, they give up easily (Skaalvik et al., 2015).As previous studies pointed out, learning motivation has a significant relationship with academic achievement. Learning motivation is helpful and constructive to academic achievement (Khalaila, 2015; Lemos & Veríssimo, 2014).

Question 9: Please explain how the results and the findings of the study will be assistive the related research community.

ANS: Thank you for your suggestion and author modify it. The contribution of the study propose an sustainable innovation learning model and use the IPMA to analyze the important design concept for virtual reality chemistry laboratory simulations game. This may pose different learning outcome because simulations and games have different design features and interactive model. The research result could provide as an experiential virtual reality learning material design guideline. (see Line 534~538)

Reviewer 3 Report

The study concerns the analyses of the students understanding about the chemical concepts and process, when a virtual chemistry laboratory is used. Several aspects were analyses like students’ self-efficacy, cognitive load, learning motivation, and academic achievements. The results were exposed in tables of values obtained from students’ questioners elaborated in classes.

The paper is too long: as diverse background is mentioned, “Earlier researches on VR”, concerning the application of VR in education; and several hypotheses H1, H2 …, were described and after worked out from the survey made over a wide group of students.

In general the study is correct, well organized and with a good analyses of results and, as a conclusion, there were carried out important indicators, encouraging the use of virtual laboratories in other areas.

Author Response

Cover Letter

Dear Editor

This is Manuscript ID sustainability-435443 major revisions of Special Issue "Selected Papers from 2nd Eurasian Conference on Educational Innovation 2019". The article title: “A Sustainability Innovation Experiential Learning Model for 3D VR Chemistry Laboratory: An Empirical Study with PLS-SEM and IPMA”.

The revisions of this paper have been clearly highlighted using the "Track Changes", and this paper has been edited by a native English speaker.

This cover letter detailing any changes as below:

Reviewer 3

The study concerns the analyses of the students understanding about the chemical concepts and process, when a virtual chemistry laboratory is used. Several aspects were analyses like students’ self-efficacy, cognitive load, learning motivation, and academic achievements. The results were exposed in tables of values obtained from students’ questioners elaborated in classes.

Question 1: The paper is too long: as diverse background is mentioned, “Earlier researches on VR”, concerning the application of VR in education; and several hypotheses H1, H2 …, were described and after worked out from the survey made over a wide group of students.

ANS: Thank you for your suggestion and author modify and rewrite it. See line 28~95.

Question 2: In general the study is correct, well organized and with a good analyses of results and, as a conclusion, there were carried out important indicators, encouraging the use of virtual laboratories in other areas.

ANS: Thank you for your suggestion and author modify it. Thank you for reviewer's appreciate. Your comments will encourage me to write more quality paper.

The Authors: Chung-Ho Su

The authors claim that none of the material in the paper has been published or is under consideration for publication elsewhere.

I am the corresponding author and my address and other information is as follows,

Address: Department of Animation and Game Design, Shu-Te University Taiwan,

+886 7 6158000 ext 6108; fax: +886 7 6158000 ext 6199.

E-mail address: [email protected]

Thank you very much for consideration!

Sincerely Yours,

Dr. Chung-Ho Su

Round  2

Reviewer 1 Report

In my opinion,  all my suggestions were accepted and followed.